# Increased Risk for Pulmonary Embolism among Patients with Ankylosing Spondylitis—Results from a Large Database Analysis

**DOI:** 10.3390/jcm13102790

**Published:** 2024-05-09

**Authors:** Omer Gendelman, Neta Simon, Niv Ben-Shabat, Yonatan Shneor Patt, Dennis McGonagle, Arnon Dov Cohen, Howard Amital, Abdulla Watad

**Affiliations:** 1Internal Medicine B, Sheba Medical Center, Tel-Hashomer, Ramat Gan 52621, Israel; omer.g79@gmail.com (O.G.); nivben7@gmail.com (N.B.-S.); yopatt123@gmail.com (Y.S.P.); howard.amital@sheba.health.gov.il (H.A.); watad.abdulla@gmail.com (A.W.); 2Faculty of Medicine, Tel Aviv University, Tel Aviv 69978, Israel; 3Leeds Institute of Rheumatic and Musculoskeletal Medicine, University of Leeds, Leeds LS9 7JT, UK; d.g.mcgonagle@leeds.ac.uk; 4Leeds Musculoskeletal Biomedical Research Centre, Chapel Allerton Hospital, Leeds LS7 4SA, UK; 5Chief Physician’s Office, Central Headquarters, Clalit Health Services, Tel Aviv 67754, Israel; arcohen@clalit.org.il; 6Siaal Research Center for Family Medicine and Primary Care, Faculty of Health Sciences, Ben-Gurion University of the Negev, Beer-Sheva 84105, Israel

**Keywords:** real world evidence, epidemiology, registries, pulmonary embolism, ankylosing spondylitis

## Abstract

**Background:** Axial spondyloarthropathy(AS) is a chronic inflammatory disease primarily affecting the axial skeleton, often characterized by sacroiliitis. While pulmonary embolism (PE), a potentially lethal condition, has been linked to several autoimmune diseases, limited data exist regarding PE risk among patients with AS. **Methods:** This retrospective cohort study utilized the Clalit Healthcare Services (CHS) database, including 5825 patients with AS and 28,356 matched controls. Follow-up began at the date of first AS diagnosis for patients and at the matched patient’s diagnosis date for controls and continued until PE diagnosis, death, or study end date. **Results:** Prevalence of PE before AS diagnosis in patients compared to controls was 0.4% vs. 0.2% (*p* < 0.01). The incidence rate of PE was 11.6 per 10,000 person-years for patients with AS and 6.8 per 10,000 person-years for controls. The adjusted hazard ratio (HR) for PE in patients with AS was 1.70 (*p* < 0.001). Subgroup analysis demonstrated excess risk for PE in patients with AS regardless of gender and age, with variations among AS treatment categories. **Discussion:** Our findings highlight a significant association between AS and PE, indicating an increased risk in patients with AS independent of age and sex and suggests a subclinical level of inflammation. Preliminary results suggest a protective role of immunosuppressing drugs. Further research into the impact of treatment strategies should be conducted and could inform clinical management and reduce the life-threatening risk of PE in Patients with AS.

## 1. Introduction

Axial spondyloarthropathy (AS) is a chronic systemic inflammatory disease that belongs to the spondyloarthropathy group [1].

The hallmark of AS is sacroiliitis, characterized by inflammatory back pain and stiffness [2].

The majority of patients manifest their first symptoms before the age of 40, and historically, AS has exhibited a male predilection [3]. However, in recent years, the gender difference has been less prominent than was previously thought [4].

AS is not solely confined to the spine; up to 36% of patients report peripheral arthritis at various stages of their disease [5], and up to 50% of patients develop inflammation at the insertion sites of tendons, ligaments, capsules, or fascia into bone, collectively known as enthesis [6]. In addition, extra-musculoskeletal manifestations, including inflammatory bowel disease (IBD), psoriasis (PSO), and anterior uveitis (AU), are frequent and contribute to the total burden of AS [7,8].

Pulmonary embolism (PE) is a potentially lethal disease, responsible for 300,000 death per year in the United States alone [9]. Several risk factors, both inherited and acquired, have been found to increase the risk of PE [10]. A variety of autoimmune diseases, including rheumatoid arthritis, lupus, and inflammatory myositis, have been associated with elevated risk for PE [11]. Our group has also previously observed a similar trend for familial Mediterranean fever (FMF), the most common autoinflammatory disease [12].

However, information regarding the risk of PE among patients with AS remains limited. To date, only a single meta-analysis incorporating data from three studies has explored this topic, reporting a pooled risk ratio of venous thromboembolism (VTE) of 1.60 (95% C.I 1.05–2.44) [13]. Therefore, the aim of this study was to assess the incidence and the risk of pulmonary embolism among patients with AS in a real-world setting by utilizing the electronic database of a large health maintenance organization (HMO) in Israel.

## 2. Methods

### 2.1. Settings

Data for this study were collected from the electronic database of Clalit Healthcare Services (CHS), the largest health maintenance organization in Israel, serving around 4.5 million insured members, accounting for over 50% of the country’s population. CHS caters to diverse ethnic groups and integrates data from pharmaceutical, medical, and administrative systems. This database serves both administrative and clinical management purposes and is accessible for epidemiological studies. Patient data can be automatically extracted using data-mining techniques. The accuracy of the CHS database was thoroughly validated, with a reported accuracy level between 90% and 100%. It has been previously utilized in numerous studies, including those involving the same ankylosing spondylitis (AS) cohort [14].

### 2.2. Sample and Design

This study was designed as a retrospective-cohort study using the CHS’s computerized database. We extracted a cohort of patients with AS first diagnosed between 1 January 2002 and 31 December 2018 and compared them with age- and sex-matched controls. For patients with AS, the follow-up period began at the date of first AS diagnosis, and for controls, it was initiated on their matched patient’s diagnosis date. Patients under the age of 18 years were excluded from further data analysis. Patients with a prior diagnosis of PE were included in the initial cohort but were excluded from the longitudinal analysis of PE incidence. Follow-up continued until diagnosis of PE, death, or end of study on 1 July 2019.

### 2.3. Study Variables

AS diagnosis was defined based on at least one documented outpatient diagnosis by either a primary care physician or specialist, or mentioned in hospital discharge records. The occurrence of PE was determined using a comparable approach. Controls were randomly selected from the CHS database, excluding patients with AS, with approximately five controls matched to each patient with AS based on age, gender, and enrollment period. Data included variables such as age, gender, socioeconomic status (SES), body mass index (BMI), chronic conditions, laboratory results, and date of death. SES was determined according to the poverty index of the member’s residential area, as outlined in the 2008 National Census, which factored in household income, educational background, marital status, car ownership, and other components. The population was categorized into three BMI groups based on quartiles: low (bottom 25%), medium (25–75%), and high (top 75–100%). BMI was calculated using height and weight measurements from the year of enrollment into the study, if available, and was categorized into four groups: underweight (<18.5 kg/m^2^), normal (18.5–24.9 kg/m^2^), overweight (25–30 kg/m^2^), and obese (>30 kg/m^2^). Patients were categorized into different treatment groups: tumor necrosis factor inhibitors (TNFi), disease-modifying anti rheumatic drugs (DMARDs), or non-steroidal anti-rheumatic drugs (NSAID) (Table 1). TNFi treatment included patients that had been prescribed and dispensed any of the following agents: infliximab, etanercept, adalimumab, golimumab, and certolizumab pegol. Patients were defined as being treated with DMARDs if they had been dispensed methotrexate or sulfasalazine. Those treated with NSAIDs were defined as any patient who had been prescribed and dispensed any NSAID agent.

### 2.4. Statistical Analysis

Baseline characteristics across different independent variables were compared, utilizing either the *t*-test or Mann–Whitney U test for continuous variables and the χ^2^ test for categorical variables. The risk for pulmonary embolism (PE) was assessed employing the Cox proportional hazards methodology and conveyed as the hazard ratio (HR) along with the corresponding 95% confidence interval (95% CI). The primary outcome was the occurrence of PE, with the independent variable being the diagnosis of ankylosing spondylitis (AS). Both univariate and multivariate models were implemented. Survival curves were constructed using the Kaplan–Meier approach, and subsequent post hoc log-rank comparisons were executed. The statistical analysis was conducted utilizing the commercial software “Statistical Package for the Social Sciences” (SPSS for Windows, V.26.0, IBM SPSS Statistics, Armonk, NY, USA).

## 3. Results

### 3.1. Study Population

Between 2002 and 2018, a total of 5825 patients with AS and 28,356 controls without AS were included in the study. Both groups had a mean age of 50 (SD ± 16), with a clear male predominance of 63.5%. The average follow-up period for patients with AS and controls was 7.35 (range: 3.4–11.4) and 7.50 (range: 3.5–11.7) years, respectively. Smoking was slightly more prevalent and statistically significant in the AS group compared to controls (34.9% vs. 33.1%, *p* < 0.001).

### 3.2. Medication Use

Naturally, only the AS cohort received medication during the study follow-up period. Most patients with AS (97%) used NSAIDs with 55.8% of patients with AS used only NSAID therapy. TNFi were used alone in 12.7% of patients with AS and 30.2% of patients used TNFi in conjunction with another medication. DMARDs were used by 29% of patients with AS and DMARDs alone were used by 11.4% of patients (Table 1).

### 3.3. Prevalence of PE

The prevalence of PE prior to the diagnosis of AS or index date for controls was significantly higher in patients with AS compared to controls (0.4% vs. 0.2%; *p* < 0.01). The mean difference between the diagnosis of AS to the diagnosis of PE was in absolute values (Table 1).

### 3.4. Incident PE Events

The median follow-up period for AS and controls was 7.35 (IQR 3.4–11.4) and 7.50 (IQR 3.5–11.7) years, respectively. During that follow-up period, 52 patients with AS (0.9%) experienced an incident PE event compared to 152 (0.5%) controls. This resulted in an incidence rate of 11.6 (95% CI 8.7–15.2) per 10,000 person-years for patients with AS, compared to 6.8 (95% CI 5.8–8.0) per 10,000 person-years for controls.

### 3.5. Risk of PE

The crude hazard ratio (HR) for PE in patients with AS was 1.70 (95% CI 1.24 to 2.33; *p* < 0.001), and this association remained significant after adjusting for age, sex, ethnicity, SES, and BMI (HR = 1.59; 95% CI 1.04 to 2.42; *p* < 0.001) (Table 2). The ancillary Kaplan–Meier survival analysis curve is presented in Figure 1.

### 3.6. Subgroup Analysis

A separate analysis was conducted for both men and women, as well as for different age groups, and the excess risk for PE in patients with AS was demonstrated in each of these analyses. Notably, when considering patients with AS according to treatment, those treated with TNFi (HR = 1.44; 0.53 to 3.91; *p* = 0.472) and those treated with DMARDs only (HR = 1.37; 0.55 to 3.38; *p* = 0.494) did not demonstrate a significant excess risk for PE. In contrast, the rest of the cohort, treated with NSAIDs only, did exhibit a significant risk for PE (HR = 1.79; 1.25 to 2.55; *p* < 0.001 (Table 3).

## 4. Discussion

This study aimed to investigate the association between ankylosing spondylitis and pulmonary embolism by utilizing a large, comprehensive database. Our findings revealed a statistically significant increase in incidence of PE among patients with AS, with an incidence rate of 11.6 per 10,000 person-years as well as a significant increase in hazard ratio for PE (1.70, *p* < 0.001). Importantly, this association remained significant after adjusting for traditional risk factors for venous thromboembolism (VTE). Very few studies have investigated the association between VTE and AS; however, our results are in line with the results of previous studies on the topic. Ramagopalan et al. [15] investigated the occurrence of VTE after hospital admission for a wide range of immune-mediated diseases, including patients diagnosed with AS. Their study looked at multiple databases in England, consisting of 976 patients with AS (30% female), 991 patients with AS (29% female), and 2001 patients with AS (29% female), and calculated the rate ratio of VTE compared with the control cohort. In contrast to other connective tissue diseases, AS is a seronegative disease. While there are similarities in increased risk of PE between AS and other connective tissue diseases, it is important to note the lack of autoantibodies in AS, especially compared to connective tissue diseases with autoantibodies known to be linked to clotting, such as in systemic lupus erythematosus [15]. The two smaller databases had nonsignificant increased rate ratios (1.35 RR, *p* = 0.26; and 1.18 RR, *p* = 0.73, respectively). When using the larger database, the study identified a significantly increased risk of VTE among patients with AS who were previously hospitalized due to their immune-mediated disease at a rate of 1.93 RR (*p* < 0.001) [14]. Similarly, Johannesdottir et al. [16] investigated the association between risk of VTE and autoimmune skin and connective tissue diseases. They found that only connective tissue disorders posed an increased risk for VTE within 90 days of diagnosis (IRR 2.3; 95% CI 1.5–3.7) and between 91–365 days of diagnosis (IRR 2.0; 95% CI 1.5–2.8). When looking specifically at patients with AS, the incidence rate ratio for developing VTE within 1 year of diagnosis was 1.6. In contrast with our findings, an increased risk for VTE risk was observed in younger individuals (IRR 2.6 (95% CI 1.7–4.0) for patients < 40 years; and IRR 1.2 (95% CI 1.0–1.4) for patients 40–69 years) compared to older individuals (IRR 1.1; 95% CI 1.0–1.2 for patients > 70 years old) [16]. Lastly, a study by Zoller et al. [11] investigated whether autoimmune disorders increase the risk of pulmonary embolism. The study followed 9498 patients with AS (27% women) for 10 years of follow-up. Similarly to the previous study, an increased risk of PE was observed in the first year after hospitalization. For patients with AS, the standardized incidence ratios (SIR) for developing PE after admission was 1.16 (95% CI 0.99–1.35). Interestingly, the relative risk declined over time from 4.27 at 1–5 years (95% CI 2.88–6.11) to 0.88 at 10 years and later (95% CI 0.67–1.14). They also observed an increased risk of PE, regardless of age or sex, which is in line with the results of our study.

While these studies are pioneers in the field, there are some disadvantages in their methodology that may affect their applicability to the general AS population. It is important to note that these studies were drawn from hospital databases and therefore may not be applicable to the general population of patients with immune-mediated disorders that do not require hospitalization. One strength of our study lies in its utilization of a large database that encompasses all patients diagnosed with AS, rather than being limited to only those who have been recently hospitalized for an immune-mediated disorder. This broader scope enhances the applicability of our findings to the entire AS patient population, including those with less severe forms of the disease.

Recent research by Aviña-Zubieta et al. [17] has shown a significantly increased risk for PE, DVT, and VTE; fully adjusted HRs (95% CI) for PE, DVT, and VTE were 1.36 (0.92 to 1.99), 1.62 (1.16 to 2.26), and 1.53 (1.16 to 2.01), respectively. Time trend analysis revealed that the highest risk was within the first year after diagnosis (hazard ratio 2.10; 95% CI 0.88–4.99), indicating a potential early association.

Several known risk factors for VTE include older age, with older adults being at higher risk compared to younger individuals and a higher risk in women compared to men, especially those over 50 years of age [18,19]. Notably, our study revealed a uniform risk for PE, irrespective of age and sex. This suggests that AS is a significant risk factor for PE, potentially neutralizing the presumed protective effect of younger age groups and male gender.

An interesting and novel finding in our study is that PE occurred more frequently prior to the diagnosis of AS, as evidenced by the higher prevalence of PE in patients with AS before their AS diagnosis compared to controls (0.4% vs. 0.2%; *p* < 0.01). While we cannot conclude that PE is a preclinical condition heralding the onset of sacroiliitis, this observation prompts consideration that patients presenting with unprovoked PE, in the setting of spondyloarthropathy features (e.g., psoriasis, inflammatory bowel disease, positive HLAB27), might have asymptomatic [20] or undiagnosed AS.

Our study also explored the impact of different treatment strategies on the risk of PE in patients with AS. Notably, the subgroup of patients with AS that were seldom treated with NSAIDs exhibited a significantly higher risk for PE (HR = 1.79 (1.25 to 2.55); *p* < 0.001), while those treated with immunosuppressing drugs, including TNFi (HR = 1.44 (0.53 to 3.91); *p* = 0.472) and DMARDs (HR = 1.37 (0.55 to 3.38); *p* = 0.494), did not demonstrate a significantly increased risk. This observation aligns with previous research by Ungpraset et al. [21], in which a large meta-analysis encompassing 21,401 VTE events found a statistically significant risk of VTE among NSAIDs users (risk ratio 1.8; 95% CI 1.28–2.52). In a similar manner, the incidence rate for VTE among RA patients was 5.15 per 1000 person-years (4.58 to 5.78) for patients treated with TNFi compared to 3.28 (3.14 to 3.43) in the general population [22]. So far, the evidence for inflammation as an etiological factor for VTE in AS is lacking; however, the protective role of TNFi and DMARDs might imply that a stronger control of inflammation can mitigate the risk for PE in patients with AS, similarly to the favorable effect of NSAIDs, DMARDs, and TNFi on atherosclerosis and cardiovascular morbidity in patients with AS [23,24].

Several limitations in our study should be acknowledged. First, the use of administrative and computerized data restricted access to more accurate clinical information, such as disease activity and extraarticular manifestations. Furthermore, potential confounders, including oral contraceptive use and hypertension [25], which increase the risk for VTE, were not available for analysis. However, our study did adjust for other risk factors such as age, sex, smoking, and BMI, enhancing the reliability of our results. Additionally, we controlled for different medications at a class level but did not segregate this to include specific drug types. 

In conclusion, our study revealed an increased risk of PE among patients with AS compared to controls. This risk remained significant after accounting for age, sex, SES, and ethnicity. The higher prevalence of PE prior to diagnosis of AS might suggest a subclinical level of inflammation that may precede musculoskeletal manifestations of AS and warrants further investigation. Future research should explore whether different treatment strategies for patients with AS can reduce the risk of developing a future PE, which may inform treatment plans in this population. The findings from this study contribute to our understanding of the complex relationship between AS and PE and shed light on potential areas for intervention and prevention.

## Figures and Tables

**Figure 1 jcm-13-02790-f001:**
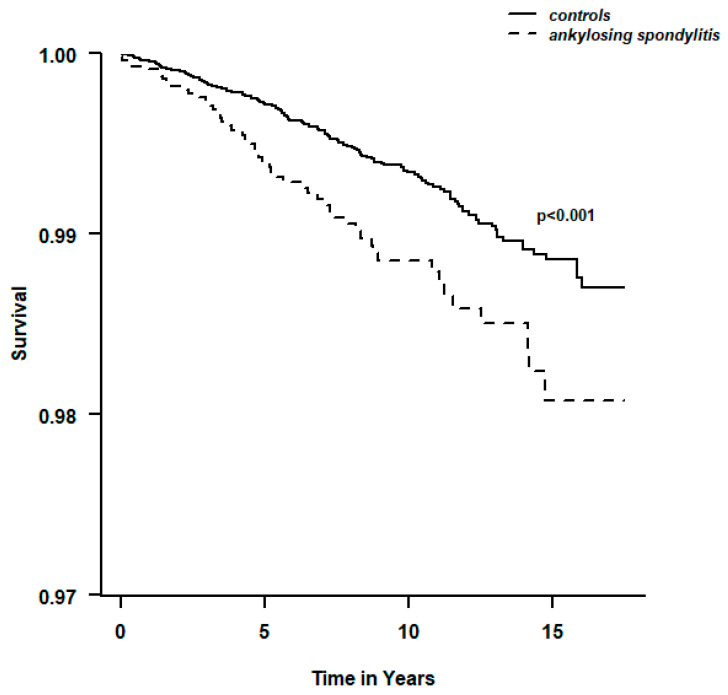
Kaplan–Meier pulmonary embolism survival plot with log-rank comparison.

**Table 1 jcm-13-02790-t001:** Baseline characteristics of the study population.

Characteristics	Patients with AS (n = 5825)	Controls (n = 28,356)	*p*-Value
Age, mean ± SD; median	50.0 ± 16; 49.3	49.8 ± 16; 49.1	NS
Men, n (%)	3701 (63.5)	17,975 (63.4.)	NS
Arab ethnicity, n (%)	999 (17.2)	4914, 17.3)	NS
Smoking history, n (%)	2032 (34.9)	9379 (33.1)	<0.01
Recruitment periods, n (%)			NS
2002–2007	1754 (30.1)	8576 (30.2)	
2008–2013	2103 (36.1)	10,212 (36.0)	
2014–2018	1968 (33.8)	9568 (33.7)	
Socioeconomic status ^a^, n (%)			NS
Low	787 (14.4)	4082 (15.4)	
Medium	3837 (70.3)	18,570 (69.9)	
High	834 (17.6)	3907 (14.7)	
Body Mass Index ^b^ n (%)			<0.001
<18.5 kg/m^2^	97 (2.3)	469 (2.5)	
18.5–24.9 kg/m^2^	1413 (33.9)	7029 (37.4)	
25–29.9 kg/m^2^	1428 (34.2)	6788 (36.1)	
≥30 kg/m^2^	1235 (29.6)	4499 (23.9)	
AS Treatment, n (%)			
Use of any TNFi	1760 (30.2)		
Use of any DMARD	1687 (29.0)		
Use of any NSAID	5648 (97.0)		
TNFi only	737 (12.7)	-	
DMARDs only	664 (11.4)	-	
Sulfasalazine only	372 (6.4)		
Methotrexate only	153 (2.6)		
Sulfasalazine and Methotrexate	550 (9.4)		
NSAIDs only	3249 (55.8)	-	
DMARDs and TNFi	1023 (17.6)	-	
TNFi and Methotrexate	664 (11.4)		
TNFi and Sulfasalazine	770 (13.2)		
TNFi, Methotrexate, and Sulfasalazine	411 (7.1)		
PE before AS/Index date	24 (0.4)	60 (0.2)	<0.01
Mean time between AS/Index to PE diagnosis, years	5.7 ± 4	6.7 ± 8	NS

Abbreviations: AS, ankylosing spondylitis; SD, standard deviation, TNFi, tumor necrosis factor inhibitors; DMARDs, disease-modifying anti-rheumatic drugs; NSAIDs, non-steroidal anti-inflammatory drugs; ^a^, available for 93.7% of data; ^b^, available for 67.1% of data.

**Table 2 jcm-13-02790-t002:** Incidence and risk for pulmonary embolism in patients with ankylosing spondylitis compared to controls.

Outcomes	Variables	AS (n = 5825)	Controls (n = 28,355)
Pulmonary Embolism	Events, n (%)	52 (0.9)	152 (0.5)
	Follow-up time, median (IQR)	7.35 (3.4–11.4)	7.50 (3.5–11.7)
	Cumulative patient’s years	44,784	222,059
	Incidence rate per 10,000 person-years, (95%CI)	11.6 (8.7–15.2)	6.8 (5.8–8.0)
	Unadjusted HR (95%CI)	1.70 (1.24 to 2.33)	reference
	Multivariate HR (95%CI)	1.59 (1.04 to 2.42)	reference

Adjusted for age, sex, ethnicity, socioeconomic status, smoking, and body mass index. Abbreviations: AS, ankylosing spondylitis; CI, confidence interval; HR, hazard ratio; IQR, interquartile range.

**Table 3 jcm-13-02790-t003:** Incidence and risk for pulmonary embolism in different ankylosing spondylitis subgroups compared to their matched controls.

Subgroup	AS n (%)	Controls n (%)	HR (95% CI)	*p*-Value
Age				
≤49 years	9 (0.3)	16 (0.1)	2.79 (1.23 to 6.31)	0.014
>49 years	43 (1.5)	136 (1.0)	1.58 (1.12 to 2.23)	0.009
Sex				
Men	31 (0.8)	88 (0.5)	1.74 (1.16 to 2.63)	0.008
Women	21 (1.0)	64 (0.6)	1.65 (1.01 to 2.70)	0.047
AS Treatment				
TNFi	5 (0.4)	17 (0.3)	1.44 (0.53 to 3.91)	0.472
DMARDs only	6 (0.7)	22 (0.5)	1.37 (0.55 to 3.38)	0.494
NSAIDs only	41 (1.2)	114 (0.7)	1.79 (1.25 to 2.55)	<0.001

Abbreviations: AS, ankylosing spondylitis; CI, confidence interval; HR, hazard ratio; TNFi, tumor necrosis factor inhibitors; DMARDs, disease modifying anti-rheumatic drugs; NSAIDs, non-steroidal anti-inflammatory drug.

## Data Availability

The data used in this study is not available upon request due to the privacy policy of CHS.

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
