# Peer review of "Increased Risk for Pulmonary Embolism among Patients with Ankylosing Spondylitis—Results from a Large Database Analysis"

_jcm, 2024, doi:10.3390/jcm13102790_

Round 1

Reviewer 1 Report

Comments and Suggestions for Authors

It was stated in the Methods that patients with prior diagnosis of PE were excluded.  However, on section 3.3 Results reports about the prevalence of prior PEs.  Can the authors pls provide clarification?  In addition, the prior diagnosis of PE was higher in the AS group, which may confound risk of PE.

This is otherwise a reasonable retrospective review of the risk of PE in patients with AS.  It is adding to the current literature, in particular in overall AS patients, rather than just using hospitalised patients.

Apart from the discordance between prior history of PE vs exclusion criteria and whether it should be included in the multivariate analysis, I do not have other concerns.  

Author Response

Thank you for your insightful comment, which has prompted us to address a crucial aspect of clarity within our study. Given the retrospective cohort study design employed, it is imperative to clarify that patients with prior PE were not excluded from the initial cohort. However, they were excluded from the longitudinal analysis of PE incidence presented in tables 2 and 3. We deemed it still pertinent to present the prevalence of prior PE in the study. Additionally, in response to your comment, we have modified the Methods section to improve clarity regarding inclusion criteria and handling of prior PE cases, aiming to enhance understanding of our study design and analysis approach.

Reviewer 2 Report

Comments and Suggestions for Authors

This is a well written and clear study of the incidence of pulmonary embolism in ankylosing spondylitis. It has the advantage of using a very large and comprehensive database which is not hospital based. It is thus a useful addition to the small number of previous studies. 

Comments:

1. In the methods it is stated that subjects with a prior history of PE were excluded. How can this be when data are presented on the prevalence of PE prior to diagnosis?

2. I don't think that AS should be casually linked to other connective diseases without some comments on their dissimilarities, principally lack of autoantibodies in AS, especially those such as seen in SLE which ae readily linked to clotting. 

3. The introduction can be shortened since it really necessary to rehearse again the standard features of AS or indeed PE.

4. The text formatting of the references needs some attention.

Author Response

Comment 1:

Thank you for your insightful comment, which has prompted us to address a crucial aspect of clarity within our study. Given the retrospective cohort study design employed, it is imperative to clarify that patients with prior PE were not excluded from the initial cohort. However, they were excluded from the longitudinal analysis of PE incidence presented in tables 2 and 3. We deemed it still pertinent to present the prevalence of prior PE in the study. Additionally, in response to your comment, we have modified the Methods section to improve clarity regarding inclusion criteria and handling of prior PE cases, aiming to enhance understanding of our study design and analysis approach.

Comment 2:

Thank you for your insightful comment, which has prompted us to further expand the discussion section of our study. In response to your comment we have added a statement regarding dissimilarities between AS as a seronegative disease and other connective tissue disorders.

Comment 3:

Thank you for drawing our attention to the length of the introduction. In line with your suggestion we have removed some background information from our introduction.

Comment 4:

Thank you for drawing our attention to the text formatting of the references. We have reformatted the spacing of this section.